# The Association of Body Composition Parameters and Simultaneously Measured Inter-Arm Systolic Blood Pressure Differences

**DOI:** 10.3390/medicina57040384

**Published:** 2021-04-16

**Authors:** Serkan Yüksel, Metin Çoksevim, Murat Meriç, Mahmut Şahin

**Affiliations:** Cardiology Department, Faculty of Medicine, Ondokuz Mayıs University, Samsun 55139, Turkey; metincoksevim@gmail.com (M.Ç.); drmeric@hotmail.com (M.M.); drmahmutsahin@gmail.com (M.Ş.)

**Keywords:** inter-arm systolic blood pressure difference, body composition parameters, visceral fat accumulation

## Abstract

*Background and Objectives*: An inter-arm systolic blood pressure difference (IASBPD) is defined as a blood pressure (BP) disparity of ≥10 mmHg between arms. IASBPDs are associated with an increased risk of cardiovascular disease (CVD). Similarly, visceral fat accumulation (VFA) is clinically important because it is associated with higher cardiovascular disease risk. Accordingly, this study compared the body composition parameters of IASBPD individuals with individuals who did not express an IASBPD. *Materials and Methods*: The analysis included 104 patients. The blood pressures of all participants were measured simultaneously in both arms using automated oscillometric devices. Then patients were divided into two groups according to their IASBPD status: Group 1 (IASBPD− (<10 mmHg)); Group 2 (IASPPD+ (≥10 mmHg)). Body composition parameters were measured using bioelectrical impedance analysis. *Results*: In 42 (40%) patients, the simultaneously measured IASBPD was equal to or higher than 10 mmHg. The right brachial SBP was higher in 63% of patients. There were no differences between the groups in terms of demographic and clinical characteristics. Regarding the two groups’ body composition parameter differences, VFA was significantly higher in group 2 (*p* = 0.014). *Conclusions*: The IASBPD is known to be associated with an increased risk of cardiovascular events. Although the body mass indexes (BMIs) of the two groups were similar, VFA levels in those with a greater than 10 mmHg IASBPD were found to be significantly higher. This finding may explain the increased cardiovascular risk in this group.

## 1. Introduction

In the guidelines for the management of hypertension, it is recommended to measure blood pressure (BP) in both arms during the first hospital visit and use the arm with a higher value for follow-up measurements [1]. As such, the inter-arm systolic blood pressure difference (IASBPD) can be evaluated, and hypertension can be better defined. From a general standpoint, IASBPD is defined as a blood pressure (BP) difference of >10 mmHg between arms and can be found in up to 24% of healthy individuals [2,3]. Available data based on angiographic and ultrasonographic imaging suggest that a ≥10 mmHg IASBPD may be a sign of subclavian and brachial arterial stenosis [4]. In addition, IASBPD is associated with subclinical atherosclerosis, left ventricular hypertrophy, aortic aneurysm/dissection, and an increased risk of cardiovascular mortality [3,5,6]. Therefore, when IASBPD is detected even in asymptomatic individuals, cardiovascular disease (CVD) risk factors may need to be managed more aggressively [7].

Obesity is an important risk factor for metabolic and CVD [8]. Body mass index (BMI), waist circumference, or waist/hip ratio can be easily measured for obesity assessment. However, it may also be clinically important to evaluate body composition parameters such as visceral fat accumulation (VFA) when performing a risk assessment. In support of this notion, VFA was found to be specifically associated with the metabolic alterations of obesity, such as metabolic syndrome or normal weight obesity, in both men and women [9,10]. Increased visceral fat accumulation was likewise found to be associated with higher cardiovascular risk and all-cause mortality [11]. Therefore, it is important to detect and treat VFA at an early stage in the general population. VFA can be measured by magnetic resonance imaging (MRI), dual-energy X-ray absorptiometry (DEXA), or computed tomography (CT) [10,12]. However, these methods have limitations (e.g., being expensive and impractical; causing radiation exposure) that are not suitable for general health checkup examinations [12]. In this context, bioelectric impedance analysis (BIA), which is a highly accessible, low-cost technique used to evaluate body composition, has been developed and validated [13,14,15]. Tetra-electrode footpad analyzers are simple and practical; thus, parameters derived from the foot-to-foot bioelectrical impedance analysis, such as body fat percentage (% BF), body muscle mass, total body water, and VFA, were compared with gold standard methods and were verified for body composition analysis [16,17].

Our study first identified the patients with an IASBPD ≥10 mmHg by simultaneously measuring blood pressures from both arms and then examined and compared the distribution of body composition parameters, specifically VFA, in patients with and without IASBPD.

## 2. Materials and Methods

The study protocol was approved by the local Ethics Committee with code number OMU-KAEK-2019-839 on 12 December 2019. All participants provided written informed consent. This study was conducted ethically in accordance with the World Medical Association Declaration of Helsinki.

One hundred and four patients (77 males and 27 females, mean ages 59 ± 12 years) who were assessed in the medical checkup clinic were included in the study. Patients with a thoracic aortic dissection, aortic aneurysm, syphilitic aortitis, aortic coarctation or Takayasu Disease, anatomic defects in both arms that would prevent blood pressure measurement, distinct edema, symptomatic heart failure (New York Heart Association class II–IV), nephrotic syndrome, and pacemaker were excluded from the study. The body composition parameters of both groups were measured using BIA and compared. Trained study nurses interviewed participants and collected data on current smoking habits, hypercholesterolemia, diabetes, and previous history of CVD. Heights and weights were measured, and BMIs were calculated. Blood samples were taken for routine evaluation.

Blood pressure (BP) measurements were performed simultaneously using two automatic oscillometric devices after a 5-min rest. The devices used for measurement have the same features (Omron M3^®^; Omron Life Science, Kyoto, Japan) and are calibrated every 6 months. Measurements were simultaneously taken at the level of the brachial artery from both arms while patients were in a sitting position. After selecting the appropriate cuff, measurements were taken from both arms twice. If a greater than 10 mmHg difference was detected in the measurements made from the same arm, a third measurement was made, and the arithmetic mean of the values was recorded.

The difference in BP between arms was expressed as the absolute difference. Absolute BP difference (| R − L |) was calculated to investigate the BP difference between the right and left arms regardless of which arm showed higher BP.

After 12 h of fasting and adherence to the pre-measurement guideline recommendations (e.g., proper hydration and a minimum resting time of 10 min before measurement), body composition parameters were measured using the bioelectrical impedance analysis method (BIA; Tanita BC601 Inner Scan^®^. Tanita Corp., Tokyo, Japan). Body composition parameter measurements were made after settling on the analyzer while standing upright with bare feet and light clothes. Physical activity states (inactive: little or no exercise; moderately active: rare, low-intensity exercise; active: regular exercises; athlete: intense exercise) other than the usual daily life activities of the participants were questioned. Height, age, gender, and physical activity information were entered in the BIA analyzer. Weight, BMI, total body fat, body muscle mass, total body water, bone mass, and visceral fat mass values were calculated automatically via the device’s unique software (Tanita Corp., Tokyo, Japan). The Tanita body composition analyzer tracks visceral fat (VF) and gives a range between 1 and 59. A rating between 1 and 12 indicates the healthy level of VF. A rating between 13 and 59 indicates the increased level of VF.

The underlying principle of the BIA method has been previously described in detail by several studies [18]. Briefly, BIA uses the body’s conductivity to measure lean mass and fat mass. Conductivity is based on the presence of free ions and electrolytes in body water. In order to measure this conductivity, an undetectable electrical current was supplied to the whole body by four surface electrodes placed in different parts of the body, and measurements were made by appropriate software [18,19].

Research data were uploaded to the computer and evaluated by “The JAMOVI project” (2020; JAMOVI 1.2 computer software]. Descriptive statistics were presented as mean ± standard deviation, frequency distribution, and percentage. Pearson Chi-Square Test and Fisher’s Exact Test were used to evaluate categorical variables. The suitability of variables to normal distribution was examined using visual (histogram and probability graphs) and analytical methods (Shapiro-Wilk Test). For the normally distributed variables, the Student’s *t*-test was used to determine potential statistical significance between the two independent groups. For those variables that were not normally distributed, the Mann-Whitney U test was used. Statistical significance level was considered a *p* < 0.05.

## 3. Results

### 3.1. Study Population

The patients included in the study were divided into two groups according to their IASBPD status (Group 1: IASBPD− (<10 mmHg), Group 2: IASBPD+ (≥10 mmHg). Sixty-two (60%) of the participants were included in group 1, and 42 (40%) were included in group 2. The basic demographic and clinical characteristics were similar between the groups. The mean BMI of the patients included in the study was 28.45 ± 5, and there was no significant difference in terms of BMI between the two groups (*p* > 0.05) (28 ± 4, vs. 28 ± 5, *p* = 0.501). Regarding laboratory measurements, the lipid profile and renal functions were similar. The fasting blood glucose value of Group 2 was significantly higher than Group 1 (113 ± 41 vs. 156 ± 87 mg/dL, *p* = 0.025). The basic demographic and laboratory features of the patients included in the study are shown in Table 1.

### 3.2. Simultaneously Measured Inter-Arm Systolic Blood Pressure Differences

The simultaneously measured IASBPD was equal to or higher than 10 mmHg in 40% of the patients. Right brachial SBP was higher in 64%, and left brachial SBP was higher in 36% of patients. The absolute inter-arm systolic, diastolic, and mean blood pressure differences of Group 2 were significantly higher than Group 1 (*p* < 0.001). Alternatively, there was no statistically significant difference between the study groups in terms of systolic, diastolic, and mean blood pressure values measured from the right and left arm (*p* > 0.05). The blood pressure measurement values of both groups are shown in Table 2.

### 3.3. Body Composition Parameters

When the body composition parameters of both groups were evaluated, the amount of visceral fat rate was higher in Group 2 (27 ± 8 vs. 31 ± 9, *p* = 0.014), while there was no statistically significant difference in other parameters. Body composition parameters of the patients included in the study are shown in Table 3.

## 4. Discussion

In this cross-sectional study, we detected that the measured VFA was higher in the IASBPD+ group; the detected FBG value was significantly higher in IASBPD+ individuals. According to these data, we expect that a high CVD risk in patients with IASBPD may be associated with increased VFA and FBG. Simply measuring VFA using the BIA method in this group can contribute to a more detailed estimation of CVD risk and to the regulation of optimal treatment.

The causes of IASBPD have not been fully elucidated; various theories include anatomical and hemodynamic descriptions and the presence of vascular obstructive disease [4,6]. IASBPD has been associated with markers of atherosclerosis, including peripheral artery disease. It is also associated with an increased risk of cardiovascular events, independent of traditional cardiovascular risk factors [3].

IASBPDs have been the subject of many studies. In a prospective study of 230 hypertensive patients (mean age 68.1 years) followed for an average of 9.8 years, 55 patients (23%) had an IASBPD ≥ 10 mmHg. In this study, the IASBPD ≥10 mmHg was associated with cardiovascular events and all-cause mortality. In addition, there was a 5–6% increase in the mortality incidence for each 1 mmHg increase in systolic blood pressure between the arms [20]. In another study, 3390 patients were followed for an average of 13.3 years (mean age 61.1 years), IASBPD > 10 mmHg was detected in 317 patients (9.4%), and a difference of ≥10 mmHg was associated with significant cardiovascular events [3]. However, although these studies provide epidemiological data, they exclude data on patients’ body composition parameters. Baseline demographic data of the patients included in our study, especially their BMI, are similar to those in the above-mentioned studies. The importance of body composition parameters is evident in determining the risk of CVD in this group of patients. Namely, individuals in the same BMI category may have variable health risks due to the amount and distribution of body fat [21,22]. In this study, we determined that VFA, which is one of the body composition parameters, is higher in patients with IASBPD.

There is controversy concerning which obesity marker is a stronger predictor of CVD [23]. In a recent study examining 6486 patients, % BF measured via BIA was independently associated with cardiovascular events [24]. However, in this study, total body fat was examined, and VFA was not evaluated. Diseases associated with obesity have been shown to be more closely related to VFA [10,14,25]. VFA has been found to be a good marker of obesity-related disorders, especially hypertension, dyslipidemia, and glucose intolerance [10,14,26]. Consistent with these data, we found high FBG in the IASBPD + group in our study. Adipocytokines secreted by VFA are thought to be effective in the pathogenic process [27]. In a study involving 257 patients, increased VFA and multiple risk factors were strongly associated with CAD [28]. Therefore, the detection of VFA in routine examinations of the general population by a simple and low-cost method is important in CVD prevention.

Excess body fat is accepted as a heterogeneous situation in which individuals with similar levels of BMI may not have similar metabolic and cardiovascular disease risks. Variations in body fat distribution may provide a potential explanation for this risk difference [29]. The white adipose tissue, which is the main component of visceral adipose tissue, secretes all of the components of renin-angiotensin-aldosteron system [21]. Additionally, under pathological conditions, adipose tissue becomes infiltrated with immune cells. Thereby resulted in an increase in proinflammatory adipokines, including tumor necrosis factor-alpha and interleukins 6 and 8, leading to insulin resistance, impaired relaxation, and vascular stiffness [27,30]. This underlying mechanism might explain both IASBPD and increased fasting blood glucose levels in patients with increased VFA. Supporting our study’s findings, Li et al. demonstrated that surrogates of visceral adiposity were strongly associated with impaired fasting glucose in non-obese Chinese individuals [31]. Similarly, the fasting plasma glucose levels were also significantly higher in patients whom increased VFA was detected.

Experimental studies show that abnormal glucose metabolism impairs normal endothelial function, accelerates atherosclerotic plaque formation, and contributes to plaque rupture and thrombosis [32]. Studies such as Rotterdam or CATHAY support these findings by determining the relationship between fasting blood glucose and CVD [33]. In our study, the high levels of FBG and VFA in the IASBPD + group demonstrate the relationship between these two risk factors.

However, only a few cross-sectional studies have evaluated associations between adiposity and IABPSD. A representative sample of 484 Finnish adults aged 25–74 years, people with IASBPD > 5 mmHg had higher BMI and arm circumference [34]. People with IASBPD showed higher BMI in the Framingham Heart Study and 806 participants aged 30–64 years without major CVD history [3]. Munoz-Torres et al. found a significant association between body fat percentage and IASBPD, but no significant associations were detected between higher quartiles of fat percentage and high IASBPD [35]. In our study, although the BMI and body fat percentages were not higher in patients with IASBPD, visceral fat levels were found to be significantly higher. This finding supports the importance of increased visceral fat level as a risk factor for IASBPD.

This study’s main limitations were small number of patients, unbalanced probands by gender and cross-sectional study design. Long-term prospective studies with higher patient numbers would provide more robust data and important additional endpoints associated with IASBPD and visceral fat accumulation.

To the best of our knowledge, this is the first study to examine the relationship between IASBPD and body composition parameters. Increased CVD risk in patients with an IASBPD ≥10 mmHg can be explained by the high amount of visceral fat. In this context, a significantly higher detection of VFA in patients with an IASBPD ≥ 10 mmHg may help to explain this situation.

## 5. Conclusions

As a result of our study, data suggesting an association between IASBPD and increased visceral fat were obtained. Visceral fat accumulation, which was easily measured by the BIA method, was significantly higher in patients with IASBPD. This finding might provide an explanation for the increased cardiovascular risk in this group based on prior studies. Thus, in the general population, the detection of visceral fat by a simple and low-cost method in routine examinations for selected individuals can contribute to the prevention of cardiovascular diseases. If the BIA methodology had a wide range of availabilities, it might be possible to use it as useful as BP measurement.

## Figures and Tables

**Table 1 medicina-57-00384-t001:** Basic Demographics, Clinical Characteristics, and Laboratory Measurements of the Patients Included in the Study.

Parameters	Group 1 *n* = 62	Group 2 *n* = 42	*p*-Value
Age (years), mean ± SD	58 ± 12	60 ± 14	0.460
Gender (F/M), *n*	15/47	12/30	0.617
BMI (kg/m^2^), mean ± SD	28 ± 4	28 ± 5	0.501
HT, *n* (%)	32 (52%)	23 (55%)	0.752
Type 2 DM, *n* (%)	19 (31%)	13 (31%)	0.973
Smoking, *n* (%)	39 (65%)	22 (52%)	0.207
Creatinine (mg/dL), mean ± SD	0.9 ± 0.2	0.9 ± 0.2	0.526
eGFR (ml/min/1.73 m^2^), mean ± SD	84 ± 19	81 ± 21	0.526
Fasting Blood Glucose (mg/dL), mean ± SD	113 ± 41	156 ± 87	0.025
Total Cholesterol (mg/dL), mean ± SD	170 ± 47	185 ± 45	0.061
Triglyceride (mg/dL), median (min-max)	150 (48–427)	132 (63–1170)	0.846
LDL-Cholesterol (mg/dL), mean ± SD	98 ± 40	108 ± 36	0.141
HDL-Cholesterol (mg/dL), mean ± SD	39 ± 12	39 ± 12	0.706

*n*: Number of patients; %: Percent, SD: Standard deviation; F/M: Female/Male; DM: diabetes mellitus; BMI: Body Mass Index; eGFR: estimated Glomerular filtration rate, HT: hypertension, LDL: low-density lipoprotein, HDL: high-density lipoprotein.

**Table 2 medicina-57-00384-t002:** Blood Pressure Measurement Values of Both Groups.

	Group 1 *n* = 62	Group 2 *n* = 42	*p*-Value
Right Brachial SBP (mmHg)	141 ± 21	148 ± 25	0.111
Right Brachial DBP (mmHg)	78 ± 12	82 ± 14	0.119
Left Brachial SBP (mmHg)	140 ± 21	141 ± 23	0.783
Left Brachial DBP (mmHg)	75 ± 12	78 ± 13	0.412
Interarm SBP difference (mmHg)	4.3 ± 2.8	17 ± 7	<0.001
Interarm DBP difference (mmHg)	5.8 ± 5.4	10.8 ± 11.9	0.002

*n*: Number of patients; SBP: Systolic blood pressure; DBP: Diastolic blood pressure.

**Table 3 medicina-57-00384-t003:** Body Composition Parameters.

	Group 1 *n* = 62	Group 2 *n* = 42	*p*-Value
Body Fat %	27 ± 9	30 ± 8	0.084
Muscle mass (kg)	54 ± 10	52 ± 11	0.290
Bone mass (kg)	2.9 ± 0.5	2.8 ± 0.6	0.292
Body water (%)	52 ±8	50 ± 6	0.082
Visceral fat rate	27 ± 8	31 ± 9	0.014

*n*: number; %: percentage.

## Data Availability

The datasets analyzed during the current study are available from the corresponding author on reasonable request.

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
