# Peer review of "The Association of Body Composition Parameters and Simultaneously Measured Inter-Arm Systolic Blood Pressure Differences"

_medicina, 2021, doi:10.3390/medicina57040384_

Round 1
Reviewer 1 Report
- small number of probands in targed groups;
- unbalanced number of probands by gender;
- it is not stated how the visceral fat rate was assessed;
- Tanita BC601 is a personal scale, the results can be considered as indicative, Tanita BC601 is not a suitable device for professional studies;
- conclusions and discussions are of a general nature;
- list of references: 22 of 33 literary sources are older than 5 years.
Reviewer 2 Report
The subject of the study is somewhat interesting but there are major concerns on conclusions the authors suggested.
Major concerns:
- The authors tried to relate the high visceral fat accumulation and difference in arm BPs, which both independently implicate increased risk of cardiovascular disease. Authors should provide more logically acceptable explanation on the relationship of visceral fat composition and increased BP difference in arms. Are these just two independent phenomena that each increase CVD risk in different mechanism? Otherwise, authors are required to provide more acceptable logical explanation on this relationship to suggest the conclusions.
2. The authors concluded that BIA methodology might be possibly used as a useful BP measurement in conclusions. There should be more acceptably logical explanation how bioelectrical impedance analysis can measure BP in discussion section to suggest this conclusion.
3. The authors may suggest how this measurement or relationship of visceral fat composition, measured by BIA, and BP difference in arms can improve or impact on patient management. The readers may benefit from any further suggestions the authors provide.
Generally, the manuscript is easy to read, and the subject is curious.
However, there is a lack of logical discussion on the link between visceral fat accumulation by bioelectrical impedance analysis and high BP difference in arms. Moreover, it is not clear how can this (any) relationship can be implicated in general population who may have increased CVD risk.
Round 2
Reviewer 2 Report
Point 1/ Response 1:
1-1) The authors should clarify the abbreviation, VAT and AT, which appeared newly in this paragraph. Or please extend full names, rather than use abbreviations.
1-2) Regarding " including TNF (tumor necrosis factor-alpha) 207
and (ILs) interleukins (IL-6 and IL-8)", I would recommend the authors minimize using too many abbreviations or use them in parenthesis following the full names.
1-3) In seventh paragraph, check "IABPD" whether it is "IASBPD".
Also, check "IASBD" whether it is "IASBPD".
2. Thank you for your revision.
Additionally,
Third paragraph in Discussion, check "IASPBD" -> "IASBPD"?
